# Revisiting One of the Dreaded Outcomes of the Current Pandemic: Pulmonary Embolism in COVID-19

**DOI:** 10.3390/medicina56120670

**Published:** 2020-12-03

**Authors:** Munish Sharma, Salim Surani

**Affiliations:** 1Corpus Christi Medical Center, Department of Pulmonary Medicine, Corpus Christi, TX 78412, USA; munishs1@hotmail.com; 2Department of Medicine, Texas A&M University, College Station, TX 77843, USA

**Keywords:** COVID-19, pulmonary embolism, anticoagulation, thrombolytic, SARS-CoV-2, DVT, thromboembolic disorder, pandemic

## Abstract

Pulmonary embolism (PE) is a commonly encountered clinical entity in patients with coronavirus disease 2019 (COVID-19). Up to 1/3 of patients have been found to have PE in the setting of COVID-19. Given the novelty of the virus causing this pandemic, it has not been easy to address diagnostic and management issues in PE. Ongoing research and publications of the scientific literature have helped in dealing with COVID-19 lately and this applies to PE as well. In this article, we attempt to succinctly yet comprehensively discuss PE in patients with COVID-19 with a review of the prevailing literature.

## 1. Background

Coronavirus disease 2019 (COVID-19), which first emerged towards the end of the year 2019, was declared a pandemic by the World Health organization (WHO) on March 11, 2020 [1]. With ever-increasing case counts across the globe, the whole world is grappling to control this pandemic. The spectrum of clinical manifestations in COVID-19 is varied, and this has added to the complexity of its management. Some patients remain asymptomatic, while a myriad of systemic complications unfortunately affect others. Among these, acute respiratory distress syndrome (ARDS) continues to be the main cause of mortality and morbidity during this pandemic [2]. Myocardial injury has been reported to significantly worsen clinical outcomes in COVID-19 patients with the mortality rate being reported to be as high as 37% in those with elevated troponin levels [3]. Besides these two entities, venous thromboembolism (VTE) due to hypercoagulability has posed substantial mortality and morbidity risks. Deep venous thrombosis (DVT) and pulmonary embolism (PE) are evident in up to 1/3 of COVID-19 patients [4]. We hereby attempt to revisit the dreaded aspect of PE in COVID-19 patients and aim to succinctly encapsulate the plethora of prevailing information in this literary composition. 

## 2. COVID-19 and Hypercoagulability

Patients suffering from COVID-19 have a hypercoagulable state, the pathogenesis of which is not completely understood [4]. Virchow’s triad that includes endothelial injury, stasis of blood, and hypercoagulable state, can be implicated with an increased incidence of VTE in COVID-19. It has been evident from a few studies that severe acute respiratory syndrome coronavirus 2 (SARS-CoV-2) directly invades endothelial cells causing injury [5]. There have been suggestions that inflammatory mediators such as interleukin-6 and various complements produced in the inflammatory cascade are also responsible for endothelial cellular injury [6,7]. In a prospective study of 33 patients, neutrophil extracellular traps (NETs) originating from decondensed chromatin that are released to immobilize the pathogens were also speculated to be one of the contributing factors in triggering immune thrombosis [8]. It is well known that venous stasis is more likely to occur with acute illness and with subsequent prolonged immobility. Sluggish blood flow through the vascular bed and impaired innate anticoagulation property during acute illness and immobility result in thrombi formation [9]. Alteration in the prothrombotic factors have been reported in patients with COVID-19. It has been reported that there is an elevation of factor VIII, fibrinogen, circulating prothrombotic microparticles, and NETs that cause hypercoagulability [10]. Hyper viscosity was demonstrated to be another possible explanation for the increased incidence of VTE in COVID-19 in a study involving 15 critically ill patients in the intensive care unit. Plasma viscosity, as determined by traditional capillary viscometry, exceeded 95% of the normal in all 15 patients. Out of these, four patients with a plasma viscosity of more than 3.5 centipoise (normal: 1.4–1.8) had thrombotic complications that included one confirmed and two suspected cases of PE [11].

## 3. Derangement in Coagulation Parameters

The effect of COVID-19 on the coagulation system can be assessed by complete blood count, prothrombin time (PT), activated partial thromboplastin time (aPTT), fibrinogen, and d-dimer values. The common abnormalities include elevated D-dimer and fibrinogen level, normal or mildly prolonged PT and aPTT, and normal, mildly reduced, or increased platelet counts [4]. There is also evidence of increased activity of factor VIII, von Willebrand factor (VWF) antigen and a small decrease in antithrombin and free protein S along with a slight increase in the protein C [4]. Thromboelastography findings suggest shortened reaction time in around 50% of patients, shortened clot formation time in around 83% of patients, increased maximum amplitude in 83% of patients and decreased clot lysis at 30 min in almost 100% of patients [4]. In a study of 64 patients admitted to the intensive care unit (ICU) in New York City hospital, thromboelastography was performed. A total of 50% of these patients were found to have a clotting index > 3, reaction time and K values were less than the normal range in 43.8% patients and maximum amplitude was more than the reference range in 60.1% patients. Patients with d-dimer > 2000 ng/mL had shorter reaction times [12]. There are also studies that have described platelet-induced tissue factor expression in monocytes. Such expression of tissue factor was found to be associated with worse severity and mortality in COVID-19 patients [13].

## 4. Comparing COVID-19 with Disseminated Intravascular Coagulopathy

Though coagulopathy in COVID-19 can manifest with an array of clinical features, the main clinical finding is related to that of thrombosis. In contrast, acute Disseminated Intravascular Coagulopathy (DIC) leads to bleeding issues predominantly. The DIC scoring system created by the International Society on Thrombosis and Haemostasis (ISTH) in 2009 [14] might hint towards a probable DIC-like state in some acutely ill patients with COVID-19 but there are some fundamental differences in the derangement of the coagulation cascade in these two entities. A DIC score of 5 or more may suggest possible DIC in acutely ill patients with COVID-19 but there has to be a clinical diagnosis as a single panel of laboratory tests would not be conclusive [14]. Coagulation panel in COVID-19 patients typically reveals high fibrinogen level and factor VIII activity in contrast to acute DIC where low fibrinogen level is the hallmark of consumption of clotting factors [4]. There is an elevation of D-dimer and mild thrombocytopenia might be evident in a few patients resembling some findings of acute DIC. Around 46% of patients in a series of 1099 were found to have an elevation of D-dimer > 0.5 mg/dl in a study in China [15,16]. Median D-dimer levels were significantly higher in patients admitted to the intensive care unit (ICU) as compared to those who did not need ICU care (2.4 mg/L vs. 0.5 mg/L, IQR 0.6–14.4) [16]. In a study conducted on 2377 adults hospitalized with COVID-19 in New York City between March and 8 April 2020, d-dimer > 2000 ng/mL had the highest risk of severe illness, acute kidney injury and thrombotic illness [17]. Prolongation of prothrombin time (PT) and activated partial thromboplastin time (aPTT) is mild in COVID-19 [15]. In a retrospective study of 183 consecutive patients in China, PT was observed to be mildly elevated in patients with severe COVID who succumbed to death (15.6 secs, Range 14.4 to 16.3 secs) as compared to patients who survived (13.6 secs, Range 13.0 to 14.3 secs) [15]. Thus, COVID-19 coagulopathy does not align itself with typical findings of acute DIC and if anything can mimic a low-grade chronic compensated DIC-like picture along with local thrombotic phenomenon that leads to organ dysfunction. 

## 5. Pulmonary Embolism in COVID-19

Clinical manifestations related to the thromboembolic phenomenon in COVID-19 have a wide spectrum and vary broadly among the patient population affected. Venous thromboembolism (VTE) includes both DVT and PE while arterial thrombosis manifests as stroke or limb ischemia,.As mentioned above, VTE that includes PE can be present in up to 1/3 of patients with COVID-19 admitted to an ICU [4]. In a study that involved postmortem examination of 21 patients who had COVID-19, PE was found in four patients. A total of 11 of these individuals were on some form of anticoagulation. They were reported to have an average age of 76 years, mean body mass index of 31 kg/m^2^, and had underlying comorbidities such as hypertension, diabetes, and cardiovascular diseases [18]. In a prospective cohort study that involved autopsies performed at an academic center in Germany, the first 12 patients who died of polymerase chain reaction confirmed COVID-19 were evaluated. 5 individuals (42%) had evidence of thrombosis in lung histology evaluation. 2 of them had D-dimer levels > 20,000 ng/mL while 1 had D-dimer > 100,000 ng/mL (normal range 100–250 ng/mL). Prophylactic anticoagulation was used in only 4 out of 12 patients. Mean BMI was 28.7 kg/m^2^ and comorbidities such as chronic kidney disease; malignancy and ulcerative colitis were most commonly noted [19]. A retrospective chart review of 3334 consecutive patients admitted from March 1 to April 17 2020 in 4 different hospitals in New York City revealed that any thrombotic event was evident in 533 (16.0%) patients. Out of these, 207 (6.2%) had VTE (3.2% PE) and 365 (11.1%) had arterial thromboembolism (1.6% ischemic stroke, 8.9% myocardial ischemia, and 1.0% systemic thromboembolism) [20]. 184 sequential patients admitted to ICU with severe COVID-19 pneumonia were studied in 2 Dutch University hospitals and 1 Dutch teaching hospital from 7 March to 5 April 2020. The total incidence of VTE was 27%, PE was reported in 14% of patients (*n* = 25) while DVT in 1. All patients were on thromboprophylaxis [21]. A multicenter prospective study was conducted in 4 hospitals in French territory between March 3 to March 31, 2020. A series of 150 ICU patients were included in the study. VTE was reported in 64 (43%) with 16.7% cases of PE. 28 patients had clotting in continuous renal replacement therapy and 2 in extracorporeal membrane oxygenation (ECMO). All patients were receiving thromboprophylaxis [22]. In a case series of 107 patients admitted to ICU in a hospital in France, a 22 patients had PE though 20 of them were already on appropriate thromboprophylaxis [23]. The incidence of PE was lower in patients not admitted to the ICU but the rate was still high enough to cause concern for patient outcome. In a French multicenter retrospective observational study that included 1240 patients who were not initially admitted to ICU, were found to have PE diagnosed by CT angiography of the chest in 103 (8.3%) patients [24]. In another retrospective, a French study of 71 non-ICU patients who were hospitalized with COVID-19 for >48 h showed PE in 10% of patients with 1 fatality. DVT occurred in 21% patients [25]. In a postmortem study of seven patients who died with severe ARDS, megakaryocyte and platelet-rich thrombi were found in the lungs. All of these patients were on anticoagulation [26] (Table 1). There are no known studies that could be currently found to determine the incidence of PE in patients not admitted to the hospital. 

## 6. Role of Diagnostic Testing

Patients in an outpatient setting do not require testing of the coagulation panel routinely unless clinically indicated [4]. In patients admitted with COVID-19, complete blood count, PT, aPTT, fibrinogen level, and d-dimer are checked at baseline and repeated daily or less frequently as per the clinical judgment of the treating physician. High D-dimer, elevated fibrinogen level along with mild thrombocytopenia, thrombocytosis, or normal platelet count are routinely observed in COVID-19 patients as discussed before [4]. Especially, the worsening of the D-dimer level has been found to indicate a worse prognosis in patients with COVID-19 [4,27]. A normal D-dimer level is considered sufficient to rule out PE in patients with low or even moderate pretest probability for PE according to the American Society of Hematology [4,28]. Patients with moderate to severe COVID-19 have been invariably found to have a higher level of D-dimer making the use of d-dimer solely to rule out PE safely a difficult task. For patients with COVID-19 who have symptoms and clinical signs suggestive of PE; cough, dyspnea, pleuritic chest pain, hemoptysis, tachycardia, hypoxia, tachypnea, evidence of DVT, Well’s score >2 and D-dimer > 500 ng/mL, more definitive testing with computed tomography angiography (CTA) chest should be done if there are no definite contraindications [29]. Many of these symptoms of the sign can be concomitantly present due to COVID-19 itself, so it might add to the complexity of clinical suspicion. In patients, who cannot undergo CTA chest, a ventilation-perfusion scan (V/Q) can be judiciously used coupled with the venous duplex scan of lower extremities. A V/Q scan may be of limited value if the patient has parenchymal changes of lungs due to COVID-19 pneumonia. For hemodynamically unstable patients, a presumptive diagnosis can be aided by bedside echocardiogram, especially if emergent life-saving treatment is being contemplated [29]. 

## 7. Role of Anticoagulants in Prevention of the PE in COVID-19

Management involves both the prevention and treatment of PE in patients with COVID-19. Unfortunately, there is no definite high-quality evidence to guide clinicians, and large clinical trials would help this cause. Meanwhile, there have been individual institutional policies and also input from various relevant societies that have helped draw a working guideline. International Society on Thrombosis and Haemostasis [30], the American Society of Hematology [31], and the American College of Cardiology [32] have proposed their recommendations and working guidelines for VTE prophylaxis as well as treatment. 

For all patients admitted to the hospital with COVID-19 (including ICU and non-ICU admissions), VTE prophylaxis should be used unless there exists a contraindication such as active bleeding. VTE prophylaxis is also recommended for peri-operative patients with COVID-19 and those who are in the hospital before or after delivery of a child [4]. Low molecular weight heparin (LMWH) is a reasonable choice if the patient is not expected to deliver within 24 h or is a post-partum status. Otherwise, unfractionated heparin (UH) would be the choice for VTE prophylaxis in obstetrics patients. LMWH/Enoxaparin at a dose of 40 mg/day can be used for creatinine clearance (CrCl) > 30 mL/min while 30 mg/day can be used for CrCl 15–30 mL/min. Alternatively, Dalteparin 5000 units daily or Nadroparin 3800 to 4000 anti-factor Xa units/day for <70 kg and 5700 units/day for >70 kg or Tinzaparin 4500 units/day can be used. If CrCl is <15 mL/min, then UH would be the preferred agent [4]. In a European study conducted in Milan, Italy, 62 patients were divided into a low-intensity care group (treated with high flow nasal cannulas), an intermediate-intensity care group (treated with continuous positive airway pressure), and a high-intensity group (requiring mechanical ventilation). The low-intensity care group received 100-units/kg/once daily of LMWH, the intermediate-intensity group received 70 units/kg twice daily and the high-intensity group received 100 units/kg twice daily LMWH. In the entire study cohort, there were thrombotic events in 25 patients that constituted 16 DVT, 8 PE, and 1 venous thrombosis in viscera [33]. The question of treatment dose anticoagulation (AC) for thromboprophylaxis has also been a topic of debate. In a retrospective study of 2773 patients hospitalized to a major hospital in New York City, 786 patients received treatment dose anticoagulation to prevent VTE-related complications. There was an improvement in the survival of intubated patients on anticoagulation as compared to those not on AC (71% vs. 37%) [34]. In another randomized trial, 20 patients were assigned to receive LMWH at 1 mg/kg subcutaneous as compared to LMWH 40 mg daily or UH 5000 units three times daily given to half the number patients each in the control group. Those on therapeutic AC were found to have a lesser number of days on a ventilator [35]. 

## 8. Treatment of Documented PE or Those with Strong Clinical Suspicion

If there is definite evidence of PE as determined by CTA chest, then initiation of full-dose anticoagulation (AC) with either LMWH 1 mg/kg subcutaneous injection every 12 h, UH based on thromboembolic nomogram or fondaparinux in case of heparin-induced thrombocytopenia (HIT) can be used unless a major contraindication exists [4]. Full-dose AC is also reasonable in patients where CTA chest or even V/Q scan is not possible, but there is strong suspicion backed by evidence of acute DVT in duplex venous ultrasound of lower extremities or clot in transit in the main pulmonary artery demonstrated by transthoracic echocardiogram. If these supportive tests are also not feasible but a patient with COVID-19 suddenly deteriorates in terms of respiratory status despite stable or normal inflammatory markers or if there is unexplained respiratory failure coupled with very high D-dimer or fibrinogen level, then use of full-dose AC would be justifiable [4]. It is extremely important to have a sound and effective follow up assessment plan formulated before discharging a patient with COVID-19 and PE. Follow-up visits are critical in terms of addressing the duration of ongoing AC, ensuring resolution of symptoms and subsequent physical and mental well-being of the patient. The choice of AC, duration, and dosing of AC should be through shared decision making with the patient. Self-monitoring of adverse effects of AC, the cost-effectiveness of the proposed medication, the importance of adherence should all be discussed explicitly before discharge. In the case of placement of inferior vena cava filter, the appropriate time for its removal should be told to the patient. Repeat echocardiogram in case of prior evidence of right heart strain, monitoring for post-PE chronic thromboembolic pulmonary hypertension, and addressing a possible post-traumatic stress disorder (PTSD) should all be included in the follow up plan [36]. It has also been recommended that facilities providing care for COVID-19 and PE join the PERT COVID-19/PE registry [36]. 

Sub-massive PE or intermediate-risk PE is not associated with hypotension, bradycardia, and does not cause cardiac arrest. However, in sub-massive PE, there are signs of right ventricular (RV) strain as evidenced by increased RV/LV (Left ventricular) ratio > 0.9 in echocardiogram or CT chest, impaired RV systolic function, elevated brain natriuretic peptide (BNP) or pro-BNP, new ST-T wave changes in the electrocardiogram and elevated Troponin I. Massive PE or high-risk PE is used to describe patients with sustained hypotension (systolic blood pressure < 90 mm Hg), bradycardia or cardiac arrest [37]. Discussion about intervention in these patients should also take into account the risks and benefits of shifting these patients to intervention suites, risk of viral transmission, and special patient circumstances. A multidisciplinary approach preferably involving Pulmonary Embolism Response Team (PERT), the largest organization in the world established to improving patient care in PE, has been recommended to be involved [4]. As per the position paper from the national PERT Consortium published in *CHEST* journal in August 2020, patients initiated on therapeutic AC who continue to deteriorate can be considered for catheter-directed thrombolysis if not deemed to be a candidate for systemic thrombolysis [4]. For patients with massive PE, it is recommended to administer systemic thrombolytic in the absence of contraindication. Failed systemic thrombolytic therapy or major contraindication to systemic thrombolysis can be addressed by catheter-directed thrombolysis [4]. The EkoSonic Endovascular System (“EKOS”) can be used for catheter-directed thrombolytics even in sub-massive PE where the evidence of RV strain and ongoing myocardial necrosis is evident. In the pre-COVID era, it has already been established that this system can effectively infuse lower doses of thrombolytic over a shorter period with good clinical outcome [4]. In the context of COVID-19, again the safety of the staff and operator in the catheterization laboratory and ensuring the safety of staff during transport of the patient should also be considered. The position paper from the PERT consortium stresses that the transfer of care for a patient with PE and COVID-19 for extracorporeal membrane oxygenation (ECMO), should be adjudicated by PERT. The decision for the safe transfer of patient or initiation of ECMO in the same institution should be made on a case-by-case basis, especially also taking into consideration the potential catheter-related thrombosis in COVID-19 [4]. There have been limited data regarding the empiric use of thrombolytic in COVID-19 patients based on clinical judgement. In a case series of four patients with severe ARDS due to COVID-19 pneumonia and shock, there was significant improvement in gas exchange and/or hemodynamics post-tissue plasminogen activator (tPA) administration. All of these patients had high d-dimer levels and had refractory hypoxemia with a significant A-a gradient. Authors have proposed that while milder cases of COVID-19 pneumonia and hypoxic respiratory failure warrant anticoagulation, more severe cases might require thrombolytic if suspicion of PE is very high [36]. A summary of the proposed treatment has been outlined in Figure 1.

There are certain challenges in the diagnosis and management of PE in COVID-19. Firstly, the clinical manifestations of acute PE itself may mimic or have commonality with acute hypoxic respiratory failure due to COVID-19 pneumonia or acute respiratory distress syndrome (ARDS). A large proportion of patients with moderate to severe COVID-19 may have acute kidney injury or worsening of chronic kidney disease making the definitive CTA chest a difficult proposition. Physicians might have to rely on clinical judgment more often than desired. Safe transfer of patients with proper use of protective personnel equipment within the hospital for imaging or catheterization laboratory and out of the hospital for transfer to a higher center can both be challenging. Patients with severe hypoxemia may not have enough reserve to counteract the burden added by a clot in the pulmonary artery ensuing RV failure and thus PE may add to the sudden fatality of COVID-19 patients. Special populations such as pregnant patients might have to endure extra caution due to the management of acute PE and AC during the peripartum period. The elderly population discharged on AC needs extra surveillance for bleeding complications [4]. In terms of continuation of anticoagulation, it is recommended that patients with documented VTE/PE require a minimum of three months of anticoagulation. Even in those patients who do not have a definite VTE or PE but are deemed to be at high risk can be eligible for post-discharge thromboprophylaxis on a case-by-case basis. Patients with a history of VTE, recent surgery and immobilization and trauma may be considered eligible after ruling out any major contraindication. Rivaroxaban 10 mg daily for 31 to 39 days has been recommended for post-discharge thromboprophylaxis based on data available so far [37]. 

## 9. Conclusions

PE in COVID-19 certainly can be a frequently encountered entity for physicians in ICU as well as non-ICU settings. Even the primary care physicians may have to be in loop to carry out the post-discharge follow up plan to prevent a multitude of potential complications. Thus, it is prudent that we understand and participate in an ever-evolving evidence-based discussion about the challenging and dreaded aspect of dealing with PE in COVID-19 patients. 

## Figures and Tables

**Figure 1 medicina-56-00670-f001:**
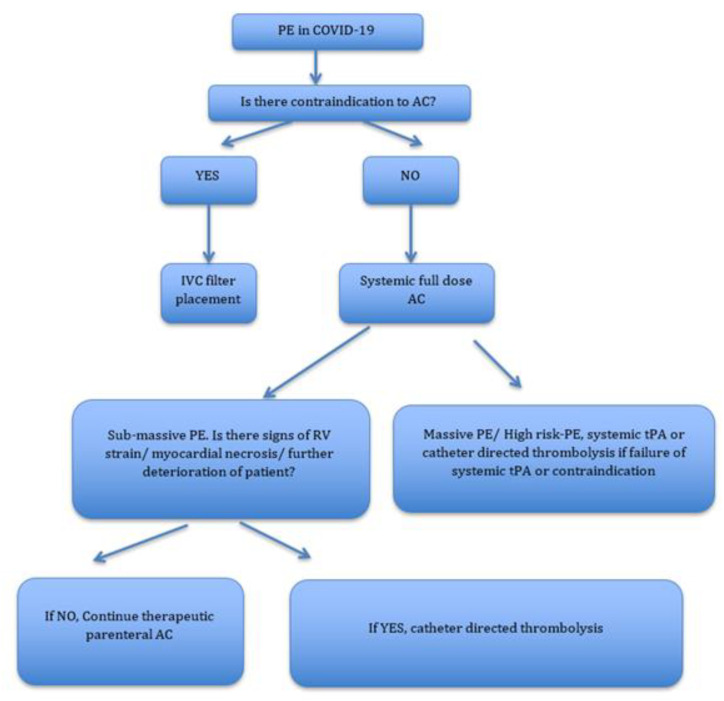
Simple flow diagram summarizing major management strategies in COVID-19 and PE. Follow up of patients with PE in COVID-19.

**Table 1 medicina-56-00670-t001:** Studies showing the incidence of pulmonary embolism (PE) in patients admitted with coronavirus disease 2019 (COVID-19) to a hospital (both intensive care unit (ICU) and non-ICU).

Study	Study Type	Major Findings
Menter T et al. (May 2020) [18]	Autopsy of 21 consecutive deceased patients. Origin of study: Basel, Switzerland.	PE found in 4 individuals, were not on anticoagulation, average age 76 years, average BMI 31 kg/m^2^, major comorbidities: hypertension, diabetes, cardiovascular diseases.
Wichmann D et al. (May 2020) [19]	A prospective cohort of 12 deceased patients. Autopsies performed. Origin of study: Hamburg, Germany.	PE in 5 individuals, only 4 out of 12 were on anticoagulation, mean age 73 years, average BMI 28.7 kg/m^2^, major comorbidities: malignancy, ulcerative colitis, chronic kidney disease (CKD).
Bilaloglu S et al. (July 2020) [20]	Retrospective chart review of 3334 patients. Origin of study: New York, USA	PE in 3.2% of patients, mean age 63 years, major comorbidities: MI, congestive heart failure, hypertension, diabetes.
Klok FA et al. (July 2020) [21]	A retrospective study of 184 patients in ICU. Origin of Study: the Netherlands.	PE in 14% of patients, mean age 64 years, mean body weight 87 kg, all patients were on thromboprophylaxis.
Helms J et al. (May 2020) [22]	A multicenter prospective cohort study of 150 patients. Origin of Study: French territory.	PE in 16.7% of patients, mean age 63 years, all patients were on thromboprophylaxis (70% prophylactic, 30% therapeutic), major comorbidities: malignancy, cardiovascular disease, diabetes.
Poissy J et al. (April 2020) [23]	Case series of 107 patients in ICU. Origin of study: Lille, France.	PE in 20.06% (*n* = 22) patients, average BMI 30 kg/m^2^ (22–53), median age 57 years (29–80), 20 of 22 patients were on prophylactic antithrombotic therapy.
Fauvel C et al. (August 2020) [24]	Retrospective observational multicenter study of 1240 patients not admitted to ICU. Origin of study: France.	PE in 8.3% patients (*n* = 103), average age 64 ± 17, BMI 28.1 ± 6.3 kg/m^2^. 837 were on anticoagulation prophylaxis. Major comorbidities: Hypertension, diabetes, hyperlipidemia, chronic kidney disease, and coronary artery disease.
Artifoni M et al. (May 2020) [25]	A retrospective study of 71 patients not admitted to ICU. Origin of study: France.	PE in 10% (*n* = 7) patients, average age 61 years (40.8–79), average BMI 27 kg/m^2^(25.5–29.1), 70 patients were on prophylactic anticoagulation. Major comorbidities: Hypertension, Diabetes, Malignancy.
Rapkiewicz AV et al. (July 2020) [26]	Autopsy case series of 7 patients. Origin of study: New York, USA.	PE in all 7 patients despite being on anticoagulation. All the thrombi were megakaryocytes and platelet rich. Average age 57.4 years, 4 females. Major comorbidities: Hypertension, Diabetes, Hyperlipidemia, obesity.

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
