# Peer review of "Revisiting One of the Dreaded Outcomes of the Current Pandemic: Pulmonary Embolism in COVID-19"

_medicina, 2020, doi:10.3390/medicina56120670_

Round 1

Reviewer 1 Report

I congratulate the authors on this very timely and well written review.  I have a few short comments. 

There is more D dimer data on risk stratification from the United States that may be helpful https://doi.org/10.1161/ATVBAHA.120.314872

Similarly there is actually a larger TEG study than the one the authors referenced. DOI: 10.1097/CCM.0000000000004471

There are some papers looking at elevated levels of platelet activation in COVID implying anti platelet agents might be useful https://doi.org/10.1182/blood.2020007252

Along with increased number of megakaryocytes in the lungs increasing platelets! https://doi.org/10.1016/j.eclinm.2020.100434

In addition,  there is actually some intriguing data about the use of TPA in COVID that would be interesting to add. https://doi.org/10.1002/ctm2.44

The PERT position paper does not exclusively recommend EKOS over other brands of catheter directed therapy and would therefore recommend removing the brand name from Figure 1. 

In addition, some sites are performing suction thrombectomy for patients with COVID and PE using either Inari Flowtriever or Penumbra CAT8 or Lightning. 

The second paragraph in the section on follow up of PE patients seems like it would be more appropriated placed in the prior section on management of acute PE. 

Author Response

Thank you for the wonderful review. We thank the reviewer.

1.) There is more D dimer data on risk stratification from the United States that may be helpful https://doi.org/10.1161/ATVBAHA.120.314872

( answer) Added data from this article. Reference 18 added as recommended by the reviewer. Highlighted in red color

2.) Similarly there is actually a larger TEG study than the one the authors referenced. DOI: 10.1097/CCM.0000000000004471

(answer) Added this study recommended by the reviewer. Incorporated data from the study as recommended above.Ref 12. Highlighted in red color.

3.) There are some papers looking at elevated levels of platelet activation in COVID implying anti platelet agents might be useful https://doi.org/10.1182/blood.2020007252

(answer) Information from this study added. Ref 13

4.) Along with increased number of megakaryocytic in the lungs increasing platelets! https://doi.org/10.1016/j.eclinm.2020.100434

(answer) Added this paper in the table and the text. Ref 28

5.) In addition,  there is actually some intriguing data about the use of TPA in COVID that would be interesting to add. https://doi.org/10.1002/ctm2.44

(answer) Added information from this paper. ref 39.

6.) The PERT position paper does not exclusively recommend EKOS over other brands of catheter directed therapy and would therefore recommend removing the brand name from Figure 1. 

(answer) removed.

7.) In addition, some sites are performing suction thrombectomy for patients with COVID and PE using either Inari Flowtriever or Penumbra CAT8 or Lightning. 

(answer) Suction/mechanical  thrombectomy in COVID patients have been described in Stroke and in rare case reports in intra cardiac cases as well. Since this manuscript is very specific to PE, not sure if including literature with specific reference would be good !

8.) The second paragraph in the section on follow up of PE patients seems like it would be more appropriated placed in the prior section on management of acute PE

(answer) As recommended by the reviewer, this paragraph has been moved in the treatment section. Entire paragraph has been highlighted by red.

Reviewer 2 Report

this is a nice review addressing Pulmunary Embolism prevention, diagnosis and treatment in patients with COVID-19.

I have only few comments 

1) would authors add some susgestions about anticoagulation in patients hospitalized and already treated with oral anticoagulants ? should they shift on heparin or not ?  

2) would authors add some information about AC treatment duration after  hospital discharge? are authors aware of any study evaluating during a longterm follow up COVID19 patients with a PE ?

Author Response

We would like to thank the reviewer for the wonderful feedback. We have addressed the reviewer's comments as described below:

1) would authors add some suggestions about anticoagulation in patients hospitalized and already treated with oral anticoagulants ? should they shift on heparin or not ? 

(answer)  Generally, if  a patient is already adequately anti coagulated on oral anticoagulants ( warfarin with therapeutic INR) or NOACs, there should not be new VTE events. We feel that home dose of anticoagulant can be continued on admission to the hospital. if a patient is not able to take oral pills, then in that case switching to a therapeutic parenteral anticoagulation would be appropriate. 

2) would authors add some information about AC treatment duration after  hospital discharge? are authors aware of any study evaluating during a longterm follow up COVID19 patients with a PE ?

(answer) Individuals with documented VTE/PE require a minimum of three months of anticoagulation. Some individuals who have not had a VTE may also warrant extended thromboprophylaxis following discharge from the hospital. Consider post-discharge thromboprophylaxis in patients with major prothrombotic risk factors such as a history of VTE or recent major surgery or trauma, as long as they are not at high bleeding risk. Options for post-discharge prophylaxis include those used in clinical trials, such as rivaroxaban 10 mg daily for 31 to 39 days ( Ref 40). 

We are not aware of a study describing long term follow up of patients with PE and COVID 19. It could be because of the emerging nature of this disease/pandemic. There might be such robust data published in future.